# Leisure-Related Social Work Interventions for Patients with Cognitive Impairment: A Systematic Review and Meta-Analysis

**DOI:** 10.3390/ijerph20031906

**Published:** 2023-01-20

**Authors:** Hui Yang, Zhezhen Lv, Yuyue Xu, Honglin Chen

**Affiliations:** 1Department of Sociology and Social Work, Minzu University of China, No. 27 Zhongguancun S St, Haidian, Beijing 100081, China; 2Department of Social Sciences, University of Eastern Finland; Yliopistonranta 1, P.O. Box 1627, FI-70211 Kuopio, Finland; 3Department of Social Work, Fudan University, No.220 Han Dan Road, Shanghai 200433, China

**Keywords:** nonpharmacological intervention, social work, cognitive impairment, systematic review, meta-analysis

## Abstract

The social work profession has been exploring nonpharmacological interventions for patients with cognitive impairment, but there are few evidence-based research outputs. Systematically evaluating the effectiveness of social work interventions for people with cognitive impairment can shed light on the matter to further improve similar interventions. Randomized controlled trials of nonpharmacological interventions for patients with cognitive impairment were selected from key literature databases in both English and Chinese from 2010 to 2021. A systematic review and meta-analysis with Revman 5.4 were performed. Seven trials were included, involving 851 patients with cognitive impairment. The meta-analysis showed that, in terms of overall cognitive function, the Montreal Cognitive Assessment score (MD = 1.64, 95% CI [0.97, 2.30], *p* < 0.001) of the intervention group was superior to the control group, but there was no significant difference in the Mini-Mental State Examination score between the two groups (MD = 0.33, 95% CI [−0.16, 0.82], *p* = 0.18). Compared with the control group, nonpharmacological intervention can effectively improve the neuropsychiatric condition of patients (SMD = −0.42, 95% CI [−0.64, −0.20], *p* = 0.0002). In summary, the current evidence shows that nonpharmacological social work interventions had a positive effect on the cognitive function and neuropsychiatric status of patients with cognitive impairment. Suggestions for future nonpharmacological intervention practice are discussed.

## 1. Introduction

Since the beginning of the 21st Century, Chinese society has experienced dramatic drops in fertility and mortality, leading to rapid increases in the average life expectancy and the rapid aging of the population. With the increased number of aging individuals, the prevalence of various chronic diseases has also increased. According to relevant data released by the International Alzheimer’s Association in 2015, there are about 9.5 million people with dementia in China, accounting for 20% of the total number worldwide. By 2030, this number is expected to exceed 16 million. At present, the number of people with dementia in China ranks first in the world, and China also has one of the fastest rates of growth in dementia in the world [1].

With the rising size of the aged population, it is expected that chronic diseases will be more common and health systems will need to pay attention to the needs of these elderly people [2]. As a greater proportion of the population survives past age 70, age-related diseases that affect daily function and mental health will impact many more people [3]. Therefore, the demand for old-age care and medical care for the elderly will inevitably increase, adding great expense to health systems [4]. A series of cognitive disorders, represented by Alzheimer’s disease, have been plaguing the elderly. According to statistics, there are 50 million dementia patients in the world, and 68% of them suffer from Alzheimer’s disease. The caregivers of these patients need to take care of them physically and emotionally, and they are under great pressure [5]. Some related studies have pointed out that the prevalence rate of MCI in the elderly over 65 years old is 15%, among which 10~15% of the people progress to AD within one year, while the incidence rate for normal people is only 1~2%, so MCI represents a group at high-risk of AD [6].

Cognitive disorder generally refers to cognitive impairment caused by various factors, covering the process from mild cognitive impairment to dementia. Cognitive dysfunction is divided into mild cognitive impairment (MCI) and comprehensive severe cognitive impairment (dementia) [7]. Because it can cause serious mental disability, cognitive disorder often is a heavy burden on families and society. The fundamental problem faced by older adults with cognitive impairment is the decline of cognitive function. The resulting emotional and life problems not only have adverse effects on their health and quality of life, but also greatly affect the physical and mental health of caregivers. Therefore, society and especially medical and social workers have sought early diagnosis and intervention for people in the cognitive impairment stage to delay the progress of the disease.

With the increasing number of patients with cognitive impairment in China, screening for and intervening in cognitive impairment among older adults has become a top concern for researchers. Nonpharmaceutical interventions have gradually become a recognized strategy to improve the cognitive health of older adults because they are more economical, convenient, and safer than traditional drug interventions. Meanwhile, nonpharmaceutical intervention has the advantages of being more cost-effective, convenient, safe, free from site restrictions, and easier to implement [8]. At present, nonpharmaceutical interventions for cognitive impairment mainly include leisure-related interventions (e.g., exercise intervention and music therapy), neural regulation technology, computer-assisted cognitive rehabilitation, acupuncture and massage, diet intervention, and nostalgia therapy [9,10]. Music therapy mainly includes listening to songs, understanding music, appreciating and learning musical instruments, and meditation, which can help with the patients’ emotions and improve their social functions; Exercise therapy normally includes walking, yoga, Tai Chi, Baduanjin, etc., which stimulate the brain cognitive function through; gardening therapy mainly stimulates patients’ “five senses” through planting flowers and plants, pruning branches and leaves, etc., achieving the effect of relaxing the body and mind; nostalgia therapy uses old photos, songs, hobbies, etc., to arouse patients’ long-term memory. In a systematic review and meta-analysis of nonpharmaceutical interventions, Tang et al. [11] confirmed the effectiveness of art therapy in reducing the symptoms of depression, anxiety, and fatigue in women with breast cancer based on a transparent literature retrieval method of systematic analysis and standardized inclusion and evaluation criteria. After systematically analyzing 22 studies on the quality of life, depression, and anxiety interventions for cancer caregivers, Fu et al. [12] showed that music therapy could ease the anxiety of caregivers and regulate their emotions.

In the process of development, social work has been constantly adopting theoretical knowledge and intervention methods from psychology. Social workers have also employed music therapy as a treatment method in cognitive impairment intervention [13] and hospice care [14] with positive outcomes. Horticulture therapy has been adopted to help empty nesters cope with loneliness [15] and older adults with chronic schizophrenia [16]. Painting therapy has been mainly used with children, adolescents [17,18], and patients with mental illness [19]. In recent years, social work practitioners have also tested nonpharmaceutical interventions for patients with cognitive impairment, using professional social work methods such as case work, group work, and group counseling [20] combined with psychological treatment modes such as memorial therapy [21] and horticultural therapy [22]. Many of these social work interventions have been implemented in combination with the previous approach, including regular exercise, diet management, etc. However, there have been few evidence-based research outputs. Evidence-based medicine involves combining clinical expertise with the best external evidence from systematic research and considering patients’ values and expectations. By combining the best external evidence with individual clinical expertise and patient preferences, doctors who practice evidence-based medicine can identify and apply the most-effective interventions to maximize their patients’ quality and length of life [23,24]. Evidence-based social work practice is based on scientific research [25], which seeks basic concepts and an operational framework of best practices from numerous sources, to understand what is and is not beneficial to the stakeholder according to the scientific research results, and to provide professional services accordingly. In evidence-based practice, the clients are active participants.

The main purpose of this study was to evaluate and meta-analyze nonpharmaceutical social work interventions for cognitive impairment by systematically investigating the methods and achievements of nonpharmaceutical interventions in the existing literature, exploring the best evidence of effective intervention for patients with cognitive impairment and establishing a scientific intervention system in terms of “practice–evidence–practice” in the field of nonpharmaceutical intervention in cognitive impairment. Accordingly, this study systematically reviewed the related research on nonpharmaceutical interventions for patients with cognitive impairment in both English and Chinese from 2010 to 2021. At the same time, a meta-analysis examined whether nonpharmaceutical social work interventions can improve the cognitive function and neuropsychiatric status of patients with cognitive impairment to provide evidence and ideas for further research on nonpharmaceutical intervention in cognitive impairment.

## 2. Materials and Methods

### 2.1. Inclusion and Exclusion Criteria

Studies using randomized controlled trials (RCTs) were included. RCTs have been mainly used in the field of medical therapy and health services, as well as drug effect detection. Evidence is the core of evidence-based medicine (EBM), and systematic reviews/meta-analyses based on RCTs are currently recognized as the highest level of evidence [26].

The inclusion criterion was patients diagnosed with cognitive impairment, from MCI to dementia. The exclusion criteria were: (a) mixed samples of participants with cognitive impairment and healthy older adults, with no separate data analysis of patients with cognitive impairment; (b) patients with cognitive impairment caused by other reasons, such as drug or alcohol dependence. The experimental groups received nonpharmaceutical interventions, whereas the control groups received a blank control or routine health education.

The main outcome was overall cognitive function. General cognitive function screening scales were adopted, including the Mini-Mental State Examination (MMSE) and Montreal Cognitive Assessment (MoCA), to investigate whether nonpharmaceutical intervention had a significant effect on the cognitive ability of patients with cognitive impairment. Secondary outcome measures included using the Neuropsychiatric Inventory to measure neuropsychiatric status, which evaluates the improvement of the neuropsychiatric and psychological aspects of patients with cognitive impairment.

Studies were excluded if they: (a) were not published in Chinese or English; (b) were duplicate publications; (c) were comments, news reports, meeting minutes, editorials; or (d) had obvious data errors, insufficient key data, or data that could not be converted into the required data format.

### 2.2. Literature Retrieval Strategy

The nonpharmaceutical intervention methods for cognitive impairment and social work practice intervention have a unified psychological basis. Therefore, to identify more comprehensive nonpharmaceutical intervention methods, this study used “social work” and “nonpharmaceutical intervention” as the search keywords.

Accordingly, the databases of the Web of Science, VIP, Wan Fang Data, and CNKI were searched by computer to collect RCTs involving patients with cognitive impairment and nonpharmaceutical interventions. The search period was from 1 January 2010 to 11 November 2021. A multiple keyword search was adopted. Chinese search terms included “dysfunction”, “neurocognitive impairment”, “dementia”, or “mild cognitive impairment in the elderly” AND “social work”, “social work intervention”, or “NPI.” English search terms included “cognitive disorder*”, “cognitive impairment”, “neurocognitive disorder*”, “AD”, “MCI”, or “dementia” AND “social work” or “NPI.”

### 2.3. Literature Screening and Data Extraction

This systematic review and meta-analysis was conducted according to the Preferred Reporting Items for Systematic Reviews and Meta-Analyses (PRISMA) 2020 (see prisma-statement.org accessed on 30 January 2022 ) checklist and flow diagram [27].

Two researchers independently screened the literature according to the inclusion and exclusion criteria, then cross-checked their findings. If there were any objections, they consulted the evaluator and discussed any issues. During the literature screening, the researchers firstly read the title and abstract imported in the Endnote library; after excluding apparently unrelated literature, further reading of the full text was performed to initially determine the inclusion of the literature, then this was assessed and discussed with the evaluators, and ultimately, the inclusion of the literature was jointly determined. Therefore, a total of 581 literatures works were excluded after excluding duplicate literature works by reviewing the title and abstract. After comprehensively reading the remaining 35 papers together with the researchers and evaluators, the two researchers independently extracted the required data, including information on the method, experimental location, number of experimental subjects, measurement tools, etc., and summarized them in an Excel sheet for a second check. If there was any missing data in the original literature, the researchers tried to contact the original author for missing information. The data were recorded as a missing value with a ”-” note in the table if there was no further information. The missing experimental data were excluded in the meta-analysis, and the reasons for the exclusion were noted in the article. In the preparation phrase of the meta-analysis, the original data were classified and summarized with the information of the experimental group, the control group, the sample size, the average value, and the standard deviation in the Excel sheet. The final analyses of the 12 studies were conducted in Revman 5.4, as shown in Figure 1.

The extracted data mainly included: (a) basic information about the study, including the first author, publication date, etc.; (b) characteristics of the research participants, such as the number of respondents in each group, their gender and age, etc.; (c) specific details of the intervention measures, including content, frequency, duration, follow-up time, etc.; and (d) outcome indicators and outcome measurement data of concern.

### 2.4. Risk of Bias Evaluation

The bias risk assessment tool for RCTs recommended by Cochrane [28] was used to assess the risk of bias in the included literature. The evaluation tool contains seven items evaluating the implementation, measurement, follow-up, and reporting bias of the included research. Specific indicators include whether the study involved a random sequence, blind method, selective reporting, etc.

### 2.5. Statistical Analysis

The meta-analysis was performed in Revman5.4. The mean difference (MD) or standardized mean difference (SMD) was used as the effect indicator, and each effect indicator included its point estimation value and a 95% confidence interval (CI). For the measurement data, this study used the difference of each outcome index variable before and after intervention as the main effect parameter and judged heterogeneity using I^2^. If there was no obvious statistical heterogeneity (I^2^ < 50%) among the research results, fixed-effect modeling was used for the meta-analysis. If there was significant statistical heterogeneity (I^2^ ≥ 50%) among the research results, random-effect modeling was used, and data with high heterogeneity were further processed by subgroup or sensitivity analysis.

## 3. Results

### 3.1. Literature Screening

The initial search identified 616 related studies; after review and screening, 12 RCTs were identified for systematic evaluation and 7 RCTs were included in the meta-analysis. See Figure 1 for the literature screening process and results.

### 3.2. Systematic Review Results

To present the research findings more clearly, the research results are summarized according to the following categories: previous research design, characteristics of research participants (age, sex, illness, etc.), evaluation scale, intervention content, intervention duration, and intervention results. The results are summarized in Table 1.

#### 3.2.1. Characteristics of Previous Studies

According to the literature from 2011 to 2021, most relevant RCTs were published after 2015, including one in 2011, one in 2014, one in 2015, five in 2018, one in 2019, and three in 2020. The research sites were diverse, including the United States, Britain, Italy, Japan, China, and Finland. Regarding the selection of the participants, three RCTs involved social screening—that is, public recruitment of patients with cognitive impairment through newspapers, the Internet, and other methods, then screening through an evaluation scale. Generally, there were conditional restrictions on recruitment. When Ngandu et al. [29] recruited patients with cognitive impairment patients, they were required to be 60 years old or older and have a score of on the Cardiovascular Risk Factors, Aging, and Incidence of Dementia scale of at least six points. The other two articles [31,34] also had corresponding requirements for participants’ age and physical condition. Other articles [38,40] identified research participants according to the pretest results of cognitive impairment scales at institutions, in communities, in apartments, and at other institutions that specialize in caring for older adults with cognitive impairment.

Regarding the experimental method, one study [29] involved a double-blind experiment, and the subjects were randomly assigned to the experimental group or the control group. Three articles [31,37,39] described single-blind experiments. The rest of the articles [30,32,33,34,35,36,37,38,40] did not specify whether a blind method was used in the experiment, but 12 studies were randomly divided into experimental and control groups. As for the measurement of the experimental results, Huang et al. [32] assessed participants after 5 months and 10 months of intervention; Yaffe [39] conducted measurements at the 6th, 12th, 18th, and 24th month of the experiment; the rest of the studies [29,30,31,33,34,35,36,37,38,40] only conducted assessments after the experiment.

#### 3.2.2. Research Participant Characteristics

The average age of participants was about 75 years old, and there were many middle-aged and older participants. The average age of participants in six studies [31,33,36,37,39,40] was between 70 and 80; the average age in two studies [32,36] was more than 80; the average age in one study [33] was younger than 70. Four articles [34,38,39,40] did not report the age of the participants. Sample sizes in each study ranged between 40 and 100 each in the experimental and control group, except for one study involving more than 500 participants in each group. More participants were women than men. Women accounted for more than 50% of the sample in six studies, including more than 70% in one study; they accounted for less than 50% in two studies, and the other four articles did not report on gender. The types of cognitive impairment studied were MCI and dementia. Six articles [30,31,32,33,37,40] reported on patients with MCI, and four articles [29,34,36,38] involved patients with dementia. In addition, one article [39] reported on patients with Alzheimer’s disease.

#### 3.2.3. Evaluation Scales

Various evaluation scales were used. To screen for cognitive impairment, the MMSE was the main assessment scale. The MMSE mainly assesses the cognitive and intellectual functions of older adults, and its contents can be divided into two parts. The first requires vocal responses only and covers orientation, memory, and attention; the second part tests the ability to name, follow verbal and written commands, write a sentence spontaneously, and copy a complex polygon similar to a Bender–Gestalt figure [41]. The MMSE was used in nine studies, with the change in the MMSE score before and after the intervention used as the primary outcome. The MoCA was used in three studies. The MoCA is a tool for rapid screening of MCI and assesses short-term memory, visuospatial abilities, executive function, attention, concentration, working memory, language, and orientation to time and place [42]. Two studies [32,36] used the Neuropsychiatric Inventory, which evaluates 10 behavioral areas: delusions, hallucination, agitation or aggression, dysphoria, anxiety, euphoria, apathy, disinhibition, irritability or lability, and aberrant motor activity [43].

Various scales were used to measure the cognitive, neurological, and psychological aspects of the participants [39]. Dimitriou [36] used the MMSE, Neuropsychiatric Inventory, German Depression Scale, and Improved Cognitive Assessment Scale in a multimodal intervention with the research participants. The German Depression Scale is a screening scale for senile depression, which measures depressed mood, reduced activity, and withdrawal pain of older adults through 30 items, and the Improved Cognitive Assessment Scale assesses the cognitive domains of attention, orientation, memory, fluency of language and words, visual memory, and visuospatial ability [36].

#### 3.2.4. Intervention Content

A comprehensive multimodal intervention was the main method in the assessed studies. Among the 12 experiments, 7 [29,30,35,36,37,39,40] used multimodal intervention—that is, two or more intervention methods were used with the experimental group. Among the multimodal interventions, exercise therapy was widely used. The seven multimodal experiments all included exercise therapy, which was combined with music therapy, aromatic massage, and cognitive training for comprehensive intervention. Online intervention is a new approach. Two experiments [29,30] provided online programs intended to intervene in the participants’ diet, exercise, and cognition. The other three studies [31,32,33] were single-intervention studies involving Wei Chi training, Tai Chi training, and cognitive behavioral therapy.

#### 3.2.5. Intervention Duration and Results

The intervention duration was also diverse, ranging from 5 days to 2 years. The longest were Ngandu’s [29] 2-year online multimodal intervention experiment for older adults with dementia and Yaffe’s [39] 2-year integrated health care for older adults with Alzheimer’s disease. The shortest intervention time was Dimitriou’s [36] multimodal intervention for older adults with dementia, which lasted 5 days. There were five studies [30,31,33,34,35] included in the article, and the intervention duration was 3 to 4 months, while other experiments [29,32,36,37,38,39,40] lasted for 8 weeks, 6 months, 10 months, or 12 months. Intervention results were mainly based on significance tests of pre- and post-intervention data. Bae’s [37] 6-month cognitive and motor intervention for older adults with MCI showed that the intervention had no significant effect by analyzing the MMSE and TMT scores. The rest of the experiments [29,30,31,32,33,34,35,36,38,39,40] were analyzed based on their respective scales, and the results showed that the intervention was effective.

### 3.3. Meta-Analysis Results

#### 3.3.1. Bias Risk Assessment of Inclusion in the Study

According to Cochrane’s bias risk assessment tool, the twelve included studies were evaluated. Overall, the quality of the included studies was relatively high, with an average of about three uncertain items. Most studies had hidden bias in the distribution. The evaluation results are shown in Figure 2.

#### 3.3.2. Included Research

Based on a systematic review of the included studies, according to the data requirements of meta-analysis, seven RCTs [30,32,33,35,36,37,40] were included in the meta-analysis, involving 851 patients with cognitive impairment. Table 2 outlines the reasons for the exclusion of the other five studies.

### 3.4. Meta-Analysis Results

#### 3.4.1. Overall Cognitive Function

In terms of overall cognitive function, five RCTs [30,32,35,37,40] were included, with nine groups of data (some articles contained multiple groups of data). Five RCTs (six groups of data) adopted the MMSE scale, I^2^ = 0, and there was no significant heterogeneity. The meta-analysis results of the fixed-effect models showed no significant difference in the MMSE scores between the experimental and control groups (MD = 0.33, 95% CI [−0.16, 0.82], *p* = 0.18; Figure 3). Two RCTs [32,33] (three groups of data) adopted the MoCA scale, I^2^ = 0, and there was no significant heterogeneity. The meta-analysis results of the fixed-effect models showed a significant difference in the MoCA scores between the experimental and control groups (MD = 1.64, 95% CI [0.97, 2.30], *p* < 0.00001; Figure 4).

#### 3.4.2. Neuropsychiatric Questionnaire

Two RCTs [32,36] (three groups of data) included the Neuropsychiatric Questionnaire, I^2^ = 41%, with no significant heterogeneity. The meta-analysis results of the fixed-effect models showed significant differences in neuropsychiatric scores between the experimental and control groups (SMD = −0.42, 95% CI [−0.64, −0.20], *p* =0.0002; Figure 5).

### 3.5. Publication Bias

A funnel plot was produced to determine the possible publication bias. The funnel diagram was extracted from 12 data results, amongst which 7 were included in the meta-analysis. As shown in Figure 6, each study showed a symmetrical distribution on both sides of the combined effect quantity, which could be considered as the study having no obvious publication bias. There were three data results outside the confidence interval, indicating that there was heterogeneity between studies. Therefore, a further sensitivity analysis was performed by excluding one study (data result) at one time to investigate the impact of the particular study on the overall risk estimation. The one-by-one exclusion found that the overall risk had no significant changes, and the results fluctuated little, ranging from 0.04 (95 % CI: −0.07–0.15) to 0.09 (95 % CI: −0.03–0.21). Therefore, the analysis results were stable and acceptable.

## 4. Conclusions

This meta-analysis included 851 patients with cognitive impairment in seven RCTs [30,32,33,35,36,37,40]. The MMSE, MoCA, and Neuropsychiatric Inventory were used as outcome indicators to investigate whether the nonpharmaceutical intervention had a significant effect on the cognitive ability and mental health of patients with cognitive impairment. The results showed that nonpharmaceutical intervention could improve the overall cognitive function of patients with cognitive impairment and was superior to the control group in relieving neuropsychiatric problems, with statistically significant differences. However, in terms of overall cognitive function, there was no significant difference between the two groups in the MMSE score, which is consistent with Mei’s [9] meta-analysis of a multicomponent nonpharmaceutical intervention for people with MCI and Zeng’s [44] meta-analysis of an exercise intervention for people with dementia. Based on this review of the relevant literature, the MMSE scale showed no significant effect, which may be due to its obvious ceiling and floor effect—that is, the sensitivity of the test may be reduced due to the favorable or unfavorable influence of personal characteristics, but not the influence of cognitive function [45]. Second, the measurement results of the MMSE are closely related to the educational level of the participants, which was not measured separately in the included RCTs. This may be one reason for the decreased sensitivity of the MMSE. In addition, some studies showed that, compared with the MMSE scale, the MoCA scale has higher sensitivity in screening for cognitive impairment, which may be related to its measurement content [46]. Besides, the participants included in this study had different stages of cognitive impairment, and the effects of nondrug intervention were also different, which may also have affected the sensitivity of the MMSE scale.

## 5. Discussion

Cognitive impairment includes the process from MCI to dementia. According to the investigation, 10–15% of patients with MCI advance to dementia every year. At present, there is no effective treatment for dementia. If intervention treatment occurs in the early stages of dementia, it may slow the progress of the disease [47]. In view of the fact that nonpharmacological intervention usually has the advantages of few adverse reactions and easy implementation, it has become a new area of interest for intervention treatment and research in recent years. Nonpharmacological intervention methods such as cognitive training, physical exercise, music therapy, and memorial therapy for patients with cognitive impairment are favored by researchers and widely used. However, according to existing research, intervention for patients with MCI mainly focuses on the improvement of cognitive ability [48]. Cognitive impairment not only leads to the decline of memory, spatial perception, attention, and task ability, but it also causes mental harm to patients, such as problems related to emotions, abnormal psychological experiences and ways of thinking, and even severe mental disorders. At the same time, the appearance of emotional apathy symptoms also increases the risk of progressing from MCI to dementia [49]. Therefore, broad intervention should be carried out for patients with MCI, including cognitive and mental aspects. Professional nonpharmacological intervention is helpful to improve the overall cognitive function of patients with cognitive impairment, whereas the improvement of neuropsychology requires family care and social support.

This research has the following implications for policy development, public health practice, and future research. The meta-analysis showed that nonpharmacological intervention has a positive effect on improving the overall cognitive function of patients with cognitive impairment. During the “14th Five-Year Plan” period in China, Alzheimer’s disease prevention and control schemes were launched, and the National Action Plan for Alzheimer’s Disease Prevention was initiated with the promotion of pilot screening for older adults’ cognitive function. It also incorporated with a massive construction of supportive infrastructures in nursing homes, care institutions, and community centers. From the policy aspect, the governmental recommendations on professional development of nonpharmaceutical interventions are promising responses for early intervention and prevention for cognitive impairment. Meanwhile, the families, communities, and nursing institutions should work jointly to provide seamless and holistic treatment mechanism for Alzheimer’s disease. In order to fulfill that, professional care personnel should be trained to support the expected services’ provision in different phases of Alzheimer’s disease prevention and intervention, namely screening, treatment, and evaluation. Nevertheless, the community should promote the public awareness of Alzheimer’s disease and facilitate early screening and nonpharmacological intervention for neighborhood residents [50]. Last but not the least, the future research should further explore the combination of various nonpharmacological intervention approaches on the basis of evidence-based research and practice, so as to explore the effective models for the treatment of the respective symptoms of cognitive impairment [51,52].

## 6. Limitations and Future Studies

The limitations of this study are as follows. First, only Chinese and English documents were searched, which is not comprehensive enough. Second, the time, frequency, intervention mode, measurement time, and outcome index of nonpharmacological interventions included in the study were heterogeneous, which may have influenced the results. Third, the included research included all stages of cognitive impairment, and the intervention effect on patients at different stages may be different, which would affect the results. Fourth, observational studies include cross-sectional studies, case-–control studies, cohort studies, etc. Compared with RCTs, observational studies have higher external authenticity. The observational studies of nonpharmacological interventions may also provide relevant data and conclusions on this topic. However, the current review excluded observational studies and only reviewed and analyzed intervention studies, which might have ignored notable information. Finally, the number of included studies was small, and the generalizability of the results is limited.

Few related studies have been conducted in China, and they were mainly based on a single intervention, lacking relevant data analysis to evaluate the intervention effect. Many related studies have occurred abroad, most of which were comprehensive nonpharmacological intervention studies. Patients with MCI and dementia have been the main subjects of intervention, with less intervention involving other types of cognitive impairment. Therefore, it is still necessary to further explore such issues as the combination of nonpharmacological interventions, the scientific measurement of intervention effects, the combination of nonpharmacological interventions being the most effective, and which nonpharmacological interventions can achieve the best results for patients at different stages of cognitive impairment. In sum, the current evidence shows that nonpharmacological intervention can significantly improve the overall cognitive function and neuropsychiatric status of patients with cognitive impairment. The early identification of and intervention in MCI is still at the forefront of research. At present, most existing achievements come from laboratory and small-sample research. How to apply these cutting-edge research results to the community of older adults who represent the greatest proportion of the aging population remains a problem in the field of early identification of and intervention in MCI [53]. Therefore, the conclusions of this study need to be further verified by large-sample and high-quality RCTs.

## Figures and Tables

**Figure 1 ijerph-20-01906-f001:**
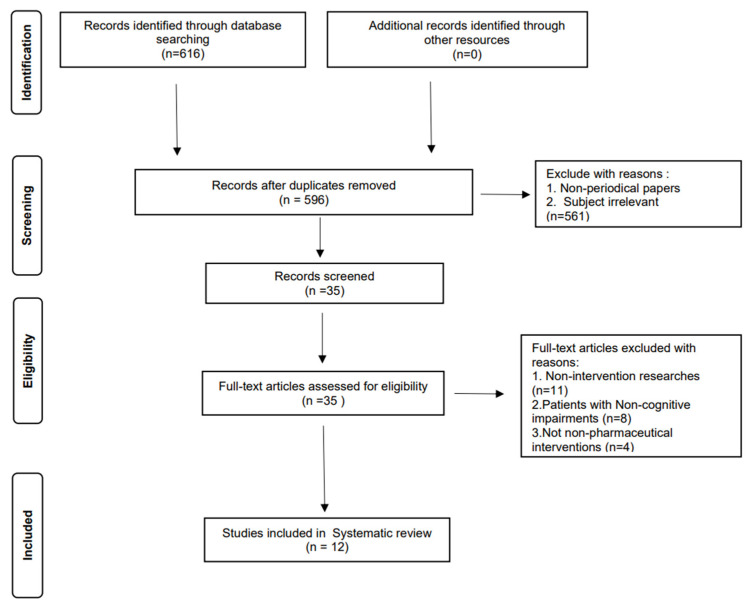
Flow diagram of this study.

**Figure 2 ijerph-20-01906-f002:**
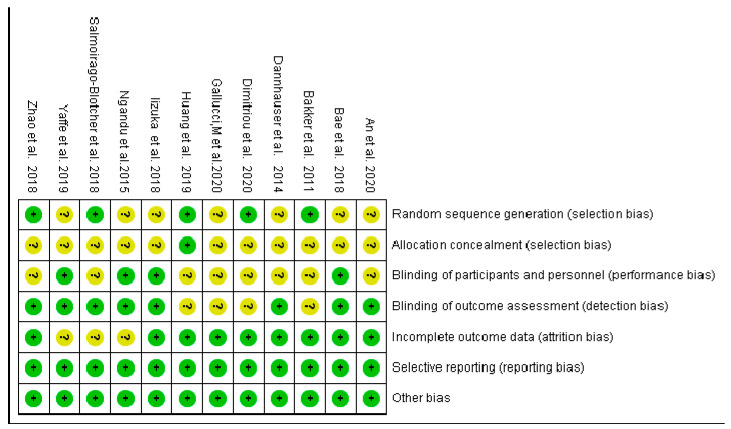
Evaluation of bias risk [29,30,31,32,33,34,35,36,37,38,39,40].

**Figure 3 ijerph-20-01906-f003:**
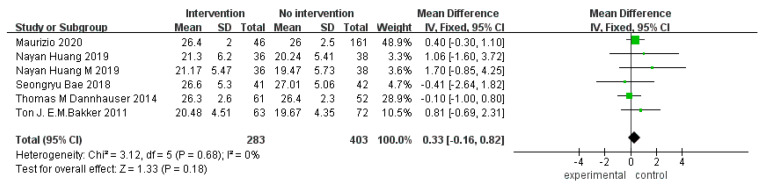
Forest plot of MMSE scale for overall cognitive function improvement [30,32,35,37,40].

**Figure 4 ijerph-20-01906-f004:**
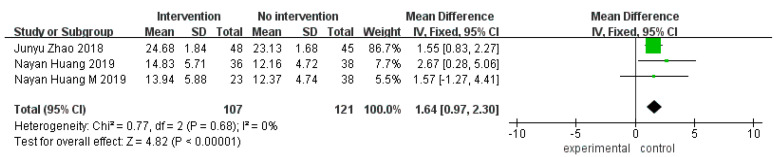
Forest plot of MoCA scale for overall cognitive function improvement [32,33].

**Figure 5 ijerph-20-01906-f005:**
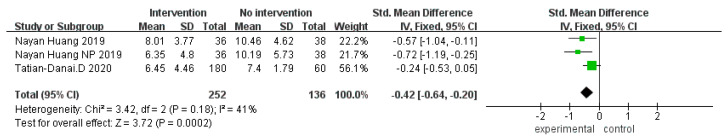
Forest plot of Neuropsychiatric Questionnaire scale for neuropsychiatric function improvement [32,36].

**Figure 6 ijerph-20-01906-f006:**
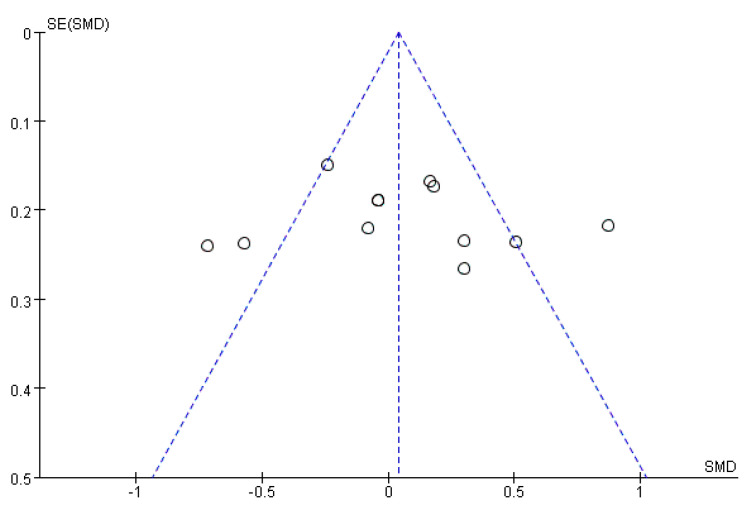
Funnel plot of interventions for Cognitive Impairment.

**Table 1 ijerph-20-01906-t001:** Basic information of included studies.

Study	Site	Experimental Method	Basic Feature of Research Participants (*n*, Sex, Age)			Scale	Content	Duration and Frequency	Results
			Experimental	Control	Disease Type				
Ngandu et al. (2015) [29]	Finland	Double-blind	591, 45% female, 69.5	599, 47% female, 69.2	Dementia	MMSE	Online training, diet, exercise, cognitive training	2 years	Notable
Dannhauser et al. (2014) [30]	Essex	--	67, 42% female, 73.8	93, 51% female, 76.4	MCI	MMSE	Online, physical activity, individual and group cognitive stimulation	12 weeks	Notable
Iizuka et al. (2018) [31]	Tokyo	Single-blind	Experiment 1: 25, 72% female, 76.8 Experiment 2: 25, 80% female, 76.5	22, 73% female, 77	MCI	MoCA, MMSE; VMST	Go training	12 weeks, 12 times	Notable
Huang et al. (2019) [32]	China	--	40, 70% female, 81.9	40, 65% female, 81.9	MCI	AVLT, MMSE, MoCA, NPI, TMT, WHO-UCLA	Tai chi	10 months	Notable
Zhao et al. (2018) [33]	China	--	48, 52.1% female, 70.6	45, 51.1% female, 69.5	MCI	MoCA	Creative expression therapy	16 weeks	Notable
Salmoirago-Blotcher et al. (2018) [34]	United States	--	30, --, --	10, --, --	Dementia	MMSE	Aerobic exercise, mindfulness training	12 weeks	Notable
Bakker et al. (2011) [35]	Rotterdam	--	54, 66.7% female, 79.8	54, 62.1% female, 81.5	Cognitive disorder	MMSE	Family therapy, life review, cognitive behavioral therapy	13 weeks	Notable
Dimitriou et al. (2020) [36]	Greece	--	60, 58.3% female, 74.7	60, 58.3% female, 74.7	Dementia	MMSE, ACE-R, GDS, FRSSD, NPI	Music therapy, transport therapy, aromatherapy massage	5 days	Notable
Bae et al. (2018) [37]	Japan	Single-blind	41, 43.9% female, 75.5	42, 52.1% female, 76.4	MCI	MMSE, TMT-A	Cognitive training, exercise	6 months, 24 times	Not notable
An et al. (2020) [38]	China	--	44;-	38, --, --	Dementia	Knowledge of dementia	Cognitive behavioral therapy	8 weeks, 8 times	Notable
Yaffe et al. (2019) [39]	United States	Single-blind	100;-	100, --, --	AD	CASI	Exercise therapy, coach intervention	2 years	Notable
Gallucci et al. (2020) [40]	Italy	--	46, 54.3% female, 74.6	161, 54% female, 76.7	MCI	MMSE	Reading therapy, motor therapy	12 months	Notable

*Note.* ACE-R = Addenbrooke’s Cognitive Examination Revised; AD = Alzheimer’s disease; AVLT = Auditory Verbal Learning Test; CASI = Cognitive Abilities Screening Instrument; FRSSD = Functional Rating Scale for Symptoms in Dementia; GDS = Geriatric Depression Scale; MCI = mild cognitive impairment; MMSE = Mini-Mental State Examination; MoCA = Montreal Cognitive Assessment; NPI = Neuropsychiatric Questionnaire; TMT = Trail Making Test; TMT-A = Trail Making Test-part A (The subjects connected 25 numbers in sequence and recorded the completion time and accuracy); WHO-UCLA = Loneliness Scale, University of California at Los Angeles; VMST = Visual Memory Span Test.

**Table 2 ijerph-20-01906-t002:** Reasons for the exclusion of studies from the meta-analysis.

Study	Reason for Exclusion
Ngandu et al. (2015) [29]	Only provided raw data of baseline MMSE measurement and lacked data after intervention
Iizuka et al. (2018) [31]	MMSE and MoCA only used as baseline measurement data, and other scales such as VMST only used in this study
Salmoirago-Blotcher et al. (2018) [34]	Missing raw data
An et al. (2020) [38]	Intervention focused on improving patients’ worries about their illness, not improving their cognitive status
Yaffe et al. (2019) [39]	Missing raw data

## Data Availability

Data used in this work can be obtained from the corresponding author upon request.

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
