# Peer review of "Leisure-Related Social Work Interventions for Patients with Cognitive Impairment: A Systematic Review and Meta-Analysis"

_ijerph, 2023, doi:10.3390/ijerph20031906_

Round 1
Reviewer 1 Report
The authors want to give a highest level of evidence about the nonpharmacological social work interventions for patients with cognitive impairment, which is very interesting for researchers focusing on mental health for older adults. However, I have a couple of comments.
1. The authors aimed to do systematic review and meta analysis for randomized controlled trial research on the nonpharmacological social work interventions for patients with cognitive impairment. However, 3/7 of the studies were not confirmed to follow randomization.
2. Test from line 132 to 162 should be moved to “introduction” section.
3. Since there some studies not listing “social work”, ”social work intervention” and ”NPI”” as keywords, but Tai Chi, Ba Duan Jin, etc. Search terms “social work”, ”social work intervention”, ”NPI” could not cover all the nonpharmacological social work intervention studies. So, the authors may omit some studies about nonpharmacological social work interventions for patients with cognitive impairment.
Author Response
Thanks for your benefitial comments. Due to a few pictures and screenshots in the answer sheet, we enclosed the saperate answer sheet in the attachment.

Reviewer 2 Report
The authors of the paper I was kindly asked to review conducted a systematic review and meta-analysis to determine the effectiveness of nonpharmacological social work interventions for people with cognitive impairment.
The paper is well structured; the methodology is clearly stated.
However, some revisions are requested in order to improve the study: details are reported below as general and specific comments.
- English revision is needed throughout the paper (i.e., repetition of sentences)
- Please, specify which version of PRISMA guidelines was used (and it should be reported in the bibliography).
- It could be helpful to specify the methods used for data extraction, including how many reviewers collected data from each paper, whether they worked independently, any processes for obtaining or confirming data from study investigators, and if applicable, details of automation tools used in the process.
- I believe authors must add a section in methods, results and discussion sections to address the risk of bias due to missing results (funnel plot and considerations about publication bias are needed) and to assess certainty in the available body of evidence.
- Please, provide a quality evaluation of all the 12 studies included in the review, not only in the metanalysis.
- I believe authors should describe the methods used to prepare the data for the quantitative synthesis and how they handled missing summary statistics or data conversions.
- Please, consider discussing more about the implications of the results for practice, policy, and future research in public health.
- Please, add details on the registration and protocol of the review, if available or state if the protocol was not registered.
- In the limitations section, it could be advisable to discuss the exclusion of observational studies from this review. They could offer more data and useful insights on the topic.
Author Response
Thanks for your kind and benefitial comments. Due to a few pictures and screenshots in the answer sheet, we enclosed the saperate answer sheet in the attachment.

Round 2
Reviewer 1 Report
1. There are two errors in Figure 1.
(1) Records after duplicates removed (n=596)→ Exclude (n=581)
596-581=15
However, the n=35 in “Records screened”
(2) The last exclusion frame should follow “Studies included in Systematic reviews”
2. Since there some studies not listing “social work”, ”social work intervention” and ”NPI”” as keywords, but Tai Chi, Ba Duan Jin, etc. Search terms “social work”, ”social work intervention”, ”NPI” could not cover all the nonpharmacological social work intervention studies. So, the authors may omit some studies about nonpharmacological social work interventions for patients with cognitive impairment, such as “Qiu Lin, Chinese Journal of Gerontology, 2017,37:3558-3560”. So I suggest that the authors should re-do literature retrieval, using Tai Chi, Ba Duan Jin, etc as search terms.
Author Response
Responses
Thanks for the supportive comments, please refer to our following answers respectively.
- There are two errors in Figure 1.
(1) Records after duplicates removed (n=596)→ Exclude (n=581)
596-581=15
However, the n=35 in “Records screened”
(2) The last exclusion frame should follow “Studies included in Systematic reviews”
Response:
Thanks, Figure 1 has been revised in the corresponding part of the article.
- Since there some studies not listing “social work”, “social work intervention” and “NPI”as keywords, but Tai Chi, Ba Duan Jin, etc. Search terms “social work”, “social work intervention”, “NPI” could not cover all the nonpharmacological social work intervention studies. So, the authors may omit some studies about nonpharmacological social work interventions for patients with cognitive impairment, such as “Qiu Lin, Chinese Journal of Gerontology, 2017,37:3558-3560”. So I suggest that the authors should re-do literature retrieval, using Tai Chi, Ba Duan Jin, etc as search terms.
Responses
Thanks for the supportive comments, please refer to our following answers respectively.
- There are two errors in Figure 1.
(1) Records after duplicates removed (n=596)→ Exclude (n=581)
596-581=15
However, the n=35 in “Records screened”
(2) The last exclusion frame should follow “Studies included in Systematic reviews”
Response:
Thanks, Figure 1 has been revised in the corresponding part.
- Since there some studies not listing “social work”, “social work intervention” and “NPI”as keywords, but Tai Chi, Ba Duan Jin, etc. Search terms “social work”, “social work intervention”, “NPI” could not cover all the nonpharmacological social work intervention studies. So, the authors may omit some studies about nonpharmacological social work interventions for patients with cognitive impairment, such as “Qiu Lin, Chinese Journal of Gerontology, 2017,37:3558-3560”. So I suggest that the authors should re-do literature retrieval, using Tai Chi, Ba Duan Jin, etc as search terms.
Response:
Thanks. In our previous revision, we reflected carefully on the reviewer’s comment that “3. Since there some studies not listing “social work”, ”social work intervention” and ”NPI”” as keywords, but Tai Chi, Ba Duan Jin, etc. Search terms “social work”, ”social work intervention”, ”NPI” could not cover all the nonpharmacological social work intervention studies. So, the authors may omit some studies about nonpharmacological social work interventions for patients with cognitive impairment.” We responded that the reason why Tai Chi, Ba Duan Jin, etc were not used as the keywords in the systematic review. We agree that there are many kinds of specific methods of nonpharmaceutical intervention, such as Tai Chi, Ba Duan Jin, etc. But not all those Tai Chi, Ba Duan Jin, etc were employed as social work interventions. Our current study focused on the social work nonpharmaceutical intervention only.
Furthermore, although Tai Chi, Ba Duan Jin, etc were not listed specifically as the keywords while doing the literature screening, studies on social work interventions with Tai Chi, Ba Duan Jin, etc approach have been included in the systematic review analysis. Therefore, we think it’s more appropriate that we don’t expand the literature review using other key words, but focus on social work nonphamaceutical intervention to remain a more concise theme of analysis. We hope our group decision could be agreeable and we are thankful for the suggestions.
As to English language, we invited a native English speaker to double check and edited the paper and hopefully it’s more fluent.
Again, thanks for your kind help.
Honglin CHEN
